# The Mechanical Role of YAP/TAZ in the Development of Diabetic Cardiomyopathy

**DOI:** 10.3390/cimb47050297

**Published:** 2025-04-23

**Authors:** Jun-Xian Shen, Ling Zhang, Huan-Huan Liu, Zhen-Ye Zhang, Ning Zhao, Jia-Bin Zhou, Ling-Ling Qian, Ru-Xing Wang

**Affiliations:** Department of Cardiology, The Affiliated Wuxi People’s Hospital of Nanjing Medical University, Wuxi People’s Hospital, Wuxi Medical Center, Nanjing Medical University, Wuxi 214023, China; junxianshen326@163.com (J.-X.S.); lingzhang07@163.com (L.Z.); hhliu1224@163.com (H.-H.L.); zhangzhenye@njmu.edu.cn (Z.-Y.Z.); zn18756929076@163.com (N.Z.); zhoujiabin2021@163.com (J.-B.Z.); qianlingling@njmu.edu.cn (L.-L.Q.)

**Keywords:** diabetic cardiomyopathy, mechanotransduction, Hippo, YAP, TAZ

## Abstract

Diabetic cardiomyopathy (DCM) begins with a subclinical stage featuring cardiac hypertrophy, fibrosis, and disrupted signaling. These changes, especially fibrosis and stiffness, often lead to clinical heart failure. The mechanism involves metabolic dysregulation, oxidative stress, and inflammation, leading to cardiac damage and dysfunction. During the progression of the disease, the myocardium senses surrounding mechanical cues, including extracellular matrix properties, tensile tension, shear stress, and pressure load, which significantly influence the pathological remodeling of the heart through mechanotransduction. At the molecular level, the mechanisms by which mechanical cues are sensed and transduced to mediate myocardial mechanical remodeling in DCM remain unclear. The mechanosensitive transcription factors YAP and TAZ fill this gap. This article reviews the latest findings of how YAP and TAZ perceive a wide range of mechanical cues, from shear stress to extracellular matrix stiffness. We focus on how these cues are relayed through the cytoskeleton to the nucleus, where they trigger downstream gene expression. Here, we review recent progress on the crucial role of YAP and TAZ mechanotransduction in the pathological changes observed in DCM, including myocardial fibrosis, hypertrophy, inflammation, mitochondrial dysfunction, and cell death.

## 1. Introduction

The global health landscape is confronted with the significant challenge of diabetes mellitus, which affected 529 million individuals in 2021 and is projected to reach a staggering 1.31 billion people worldwide by 2050 [1]. Moreover, patients diagnosed with diabetes mellitus exhibit a markedly increased risk of developing heart failure (HF). Specifically, this risk is more than twofold higher for both reduced ejection fraction (HFrEF) and preserved ejection fraction (HFpEF) compared to non-diabetic individuals [2].

Historically, diabetic cardiomyopathy (DCM) has been recognized as a cardiac disorder attributed to insulin resistance, hyperinsulinemia, and hyperglycemia. Independent of factors like dyslipidemia, hypertension, and coronary artery disease, it is initially manifested by abnormalities in diastolic relaxation, ultimately progressing to clinical HF [3]. Diabetes mellitus seldom operates as the sole determinant of myocardial dysfunction; rather, its interaction with obesity, arterial hypertension, chronic kidney disease (CKD), and/or coronary artery disease (CAD) frequently precipitates additive myocardial injury. In light of this, the Heart Failure Association of the European Society of Cardiology (ESC) has broadened its definition of DCM to include left ventricular (LV) systolic and/or diastolic dysfunction in individuals with diabetes mellitus [4]. In the initial phases, DCM encompasses a latent subclinical stage, distinguished by structural and functional anomalies, encompassing LV hypertrophy, fibrosis, and disrupted cell signaling pathways. These pathophysiological alterations, specifically cardiac fibrosis and increased stiffness, along with the accompanying subclinical diastolic impairment, frequently progress to HFpEF and ultimately culminate in systolic dysfunction, manifested as HFrEF [5].

Clinically, these processes manifest as elevated cardiac biomarkers, impaired myocardial function, and increased filling pressures, ultimately leading to overt HF [6]. Mechanistically, DCM is characterized by cardiomyocyte hypertrophy, interstitial inflammation, fibrosis, and potential cardiovascular autonomic neuropathy. Its pathogenesis involves metabolic derangements, mitochondrial dysfunction, oxidative stress, reduced nitric oxide, advanced glycation products, impaired calcium handling, inflammation, neuroendocrine activation, endoplasmic reticulum stress, and microvascular abnormalities [5]. Among them, the increased stiffness of cardiomyocytes and extracellular matrix (ECM) resulting from advanced glycation end products (AGEs) and collagen deposition leads to abnormal mechanical transduction in the heart, further exacerbating the structural and electrical remodeling of the heart.

These mechanical stimuli can be converted into biochemical signals through a process termed mechanotransduction. The processes of mechanosensing and mechanotransduction in cardiomyocytes are intricate, as they not only respond to external mechanical forces but also generate internal mechanical forces. Studies have demonstrated that cardiomyocytes are capable of responding to various mechanical stimuli, encompassing static and dynamic stresses, compressive and tensile stresses, as well as shear stress and ECM stiffness [7]. The Hippo pathway serves as an integrator of cellular responses to various mechanical stimuli, including tension, stretching, and changes in ECM properties. It comprises multiple robust cellular feedback loops that ensure cellular homeostasis. The effector Yes-associated protein (YAP) and transcriptional co-activator with PDZ binding motif (TAZ) within this pathway function as a mechanotransduction switch, responding to diverse mechanical forces, such as shear stress, cell shape alterations and ECM stiffness, and converting these into cell-specific transcriptional responses [8]. Notably, the Hippo-YAP signaling pathway, which modulates apoptosis and cell proliferation, plays a pivotal role in cardiac development, growth, homeostasis, disease progression, and regeneration. Its activation has been implicated in heart conditions such as ischemia/reperfusion injury, cardiac remodeling, and HF [9].

Although some studies have explored the role of mechanotransduction in the regulation of cardiovascular diseases, the regulatory mechanisms of mechanical signals in DCM remain poorly understood. Given the significant role of the Hippo-YAP pathway in mechanotransduction, this review focuses on the alterations in mechanical signals in DCM and the corresponding changes in YAP/TAZ on the setting of such mechanical stimuli. Furthermore, we attempt to elucidate the potential relationship between YAP/TAZ and the major pathological processes of DCM, including myocardial fibrosis, cardiac hypertrophy, oxidative stress, inflammatory response, and programmed cell death, with the aim of revealing possible therapeutic targets and identifying potential areas of research.

## 2. Characteristics of the Hippo/YAP Pathway

### 2.1. Composition and Intricate Regulation of the Hippo/YAP Pathway

The Hippo pathway, with its highly conserved key compositions, plays a pivotal role in regulating cell proliferation, apoptosis, and organ size control [10]. The pathway is composed of a series of core components, including Mammalian Ste20-like protein kinases 1/2 (MST1/2, alias STK4/STK3), Salvador family WW domain-containing protein 1 (SAV1, also termed WW45), Large tumor suppressor kinases 1/2 (LATS1/2), MOB kinase activator 1A/B (MOB1a/b), and the transcriptional co-activator Yap/TAZ [11]. Upon activation, the activation of MST1/2, in conjunction with the adaptive SAV1 protein, facilitates the capture and phosphorylation of LATS1/2 kinases. Subsequently, the phosphorylation of LATS1/2, along with the binding of MOB1 a/b, inactivates the major effectors of the Hippo pathway, namely the transcription co-activators YAP and TAZ. This mechanism exemplifies the negative regulation of YAP and TAZ activity, whereby MST1/2 and LATS1/2 promote their cytoplasmic retention and subsequent degradation [12]. In summary, the Hippo core module’s phosphorylation cascade inhibits YAP/TAZ-driven transcription by promoting nuclear exit, cytoplasmic retention, and proteasomal degradation. Conversely, when the Hippo pathway is inactivated, its MST1/2 and LATS1/2 kinases remain inactive, permitting unphosphorylated YAP and TAZ to translocate to the nucleus. In the absence of an active regulatory module or upon exposure to stimuli that activate YAP/TAZ independently of Hippo kinases, these transducers accumulate in the nucleus, where they interact with transcriptional partners to mediate the transcription of target genes (Figure 1) [13]. Mechanotransduction cues and signals transduced by ECM-binding integrins inactivate LATS1/2 to trigger YAP/TAZ dephosphorylation [9]. This process ultimately results in the nuclear accumulation of hypo-phosphorylated YAP/TAZ, which subsequently binds to transcription factors, predominantly TEA domain transcription factor family (TEADs), regulating the expression of genes implicated in cell proliferation, survival, and various other cellular processes.

### 2.2. YAP/TAZ: The Molecular Effectors and Mechanic-Transducers of the Hippo Pathway

Furthermore, YAP serves as a transcriptional co-activator, possessing a transcription activation domain but lacking a DNA binding domain. It is regarded as an analog of TAZ, with their functions often considered redundant and collectively referred to as YAP/TAZ [8]. YAP/TAZ exerts its functions through cytoplasmic-nuclear shuttling, with the terminal physiological output of the Hippo kinase cascade aimed at restraining YAP/TAZ transcriptional activity by impeding their nuclear entry. The regulatory mechanisms governing YAP and TAZ are multifaceted, incorporating cell polarity, adhesion regulators, G protein-coupled receptors, mechanical cues, and ECM-derived signals [14]. Notably, recent advancements have elucidated that cellular mechanotransduction serves as a predominant determinant of YAP and TAZ regulation, primarily functioning in a LATS-independent manner [8]. Furthermore, YAP and TAZ decode a diverse range of biomechanical signals and convert them into biological effects specific to cell types and mechanical stresses, thereby augmenting the intricacy of their regulatory mechanisms. Consequently, the identification of YAP and TAZ as mechanotransducers has elucidated the role of aberrant cell mechanics in cardiovascular diseases, encompassing atherosclerosis, fibrosis, myocardial stiffness and cardiac hypertrophy [9].

## 3. Mechanical Cues in Diabetic Cardiomyopathy

Mechanical cues in the microenvironment, including ECM properties, stretch-mediated mechanics, shear stress, and flow-induced hemodynamics, exert a significant influence on regulating vascular morphogenesis and cardiac remodeling through mechanotransduction [9]. However, these aberrant mechanical signals do not typically emerge spontaneously; rather, they often manifest in the preclinical stages of disease. For instance, in the early stage of DCM, diabetes mellitus alters the nanomechanical properties of cardiomyocytes, resulting in increased cellular and LV stiffness. Studies using atomic force microscopy compared diabetic and normal cardiomyocytes, revealing significantly higher stiffness in diabetic cells [15]. The progression of DCM is characterized by increased cardiac tissue stiffness, which collaboratively intensifies myocardial cell injury under high glucose conditions [16]. The increase in cardiac tissue stiffness may be associated with abnormalities in both the quantity and quality of the ECM within the heart. A 5-year prospective study employing cardiac magnetic resonance (CMR) and molecular analysis has revealed that the decline in contraction dynamics of DCM is consistent with the deterioration of myocardial viscoelastic properties, suggesting the involvement of the cardiac ECM [17]. Furthermore, CMR T1 mapping quantifies ECM expansion in early DCM, highlighting an association between increased ECM and LV diastolic dysfunction [18].

Distinct hemodynamic alterations are observed within the mechanical microenvironment of DCM. Specifically, hemodynamic profiles of HF patients with diabetes mellitus deviate from those of non-diabetic individuals, potentially impacting clinical outcomes. Notably, patients with diabetes mellitus, particularly those exhibiting suboptimal glycemic control, demonstrate elevated filling pressures as a notable hemodynamic parameter [19]. Furthermore, the diabetic population frequently experiences chronic pressure overload (PO), exacerbating myocardial dysfunction. Indeed, type 2 diabetes mellitus patients exhibit higher 24-h systolic blood pressure, which is independently associated with concentric LV remodeling, diastolic dysfunction, and reduced myocardial perfusion reserve [20]. In high-fat diet (HFD)-fed mice, PO induces LV dysfunction, accompanied by cardiomyocyte dedifferentiation. Remarkably, transverse aortic constriction (TAC) accelerates the progression of LV dysfunction in HFD-fed mice compared to those on a normal diet, as demonstrated by reduced fractional shortening, increased LV end-diastolic dimension, and elevated mortality [21]. Additionally, DCM exhibits significant contractile dysfunction, characterized by increased LV pressure (LVP) and elevated wall tension [22].

It has been established that mechanical stimulation via hypertensive pressure alone is sufficient to induce a phenotypic switch of arterial vascular smooth muscle cells (VSMCs), ultimately promoting the retention of a higher proportion of cell-stored lipid droplets, which subsequently triggers apoptosis, secondary necrosis, and inflammatory response [23].

In addition to tissue stiffness and pressure overload, a hallmark of DCM is the impairment of myocardial blood flow, attributable to diabetes-induced reduction in myocardial capillary density and impairment of maximal coronary blood flow [24]. Nonetheless, the mechanosensing of diverse signals, encompassing pressure, stretch, stiffness, and flow, is likely intricate and intertwined, involving an array of additional receptors, transmitter molecules, and effector molecules. In the context of DCM, the morphology and structure of the heart undergo alterations, accompanied by the emergence of myocardial anisotropy, localized changes in tissue pressure, and activation of mechanical signaling pathways. These changes subsequently lead to site-specific modifications in YAP expression, nuclear translocation, and signaling (Figure 2).

## 4. YAP/TAZ Mechanotransduction

Aberrant mechanical signaling, arising from alterations in the physical and structural characteristics of the cellular microenvironment or defects in cellular mechanosensation, has been implicated in the pathogenesis of numerous diseases. Notably, a significant proportion of these defects ultimately exert their influence on cell behavior by deregulating YAP and TAZ [8]. Mechanical forces, including extracellular matrix, hydrostatic pressure, compressive force, tension, shear stress, and pathological stretch, regulate cellular behavior. These forces modulate the YAP/TAZ signaling pathway, through which YAP/TAZ activation and nuclear translocation mediate the cellular response to mechanical signals and their impact on the surrounding microenvironment.

### 4.1. Extracellular Matrix Properties

The mechanical microenvironment of cells is largely determined by their neighbor cells and the surrounding ECM. Mechanical cues emanating from this extracellular microenvironment, including the elasticity, viscosity, and viscoelasticity of the ECM, play a crucial role in regulating cellular behaviors such as migration, proliferation, and differentiation. This regulation occurs through the modulation of YAP/TAZ translocation dynamics [25]. Furthermore, this process exhibits a self-reinforcing nature. Specifically, increased ECM stiffness [26], elevated viscosity [27], and enhanced viscoelasticity [28] collectively enhance integrin binding. This, in turn, facilitates cytoskeletal contraction, leads to nuclear compression, opens nuclear pore channels, and ultimately enhances the nuclear accumulation of YAP/TAZ. In response to the elevated stiffness of the substrate and collagen stimulation, YAP/TAZ translocates to the nucleus and subsequently binds to chromatin [29]. Moreover, enhanced YAP activation stimulates ECM deposition by increasing the secretion of connective tissue growth factor. This, in turn, promotes the synthesis of fibronectin and collagen by surrounding fibroblasts [30]. Additionally, in cultured mesangial cells (MCs), high glucose directly activates YAP/TAZ via the canonical Hippo pathway. This hyperactivation mimics the ECM deposition observed in MCs from mouse models [31].

### 4.2. Pressure Overload and Hydrostatic Pressure

Pressure overload (PO), which arises from hypertension or aortic valve stenosis, induces cardiac hypertrophy as a compensatory mechanism to maintain wall stress and cardiac output; however, this ultimately results in cardiac failure, characterized by increased cardiomyocyte death. Although the activation of YAP promotes cardiomyocyte regeneration after cardiac injury, its prolonged activation under PO conditions triggers cardiomyocyte dedifferentiation and HF through the activation of the YAP-TEAD1 mechanism [32]. Importantly, YAP, persistently activated in diabetic hearts, plays a pivotal role in mediating the exacerbation of HF in response to PO in the hearts of mice maintained on an HFD [21].

Blood hydrostatic pressure, a significant determinant of the endothelial biophysical environment, exerts isotropic compressive stress on cells. This mechanical stress activates YAP and initiates the signaling, potentially by facilitating YAP nuclear translocation, inducing its transcription, or both [33]. Furthermore, elevated oscillating hydrostatic pressure activates YAP/TAZ, which in turn induces TEAD-mediated transcription, leading to disruption of ECM homeostasis and intensification of cell apoptosis [34,35].

### 4.3. Shear Stress

Shear stress, representing a fluid frictional force, constitutes another significant mechanical stimulus that sustains tissue homeostasis. Indeed, endothelial cells (ECs) lining the heart chambers and blood vessels are continuously subjected to shear stress from blood flow, and they are capable of perceiving and responding to changes in flow direction and magnitude of shear stress through mechanical sensors and mechanosensitive signaling pathways. Proper shear stress sustains endothelial homeostasis, whereas disturbed shear stress within the fibrotic microvascular system stimulates endothelial dysfunction. Exposure of cultured ECs to atheroprone disturbed flow triggers YAP/TAZ activation and subsequent nuclear translocation, leading to the upregulation of target genes that promote EC proliferation and inflammation [36]. Conversely, atheroprotective unidirectional shear stress exerts an inhibitory effect on YAP/TAZ activity [37]. Shear stress or disturbed flow upregulates YAP/TAZ, promoting glycolytic metabolism and inflammation, which are inhibited by statin treatment [36,37]. In microfluidic channels, wall shear stress triggers the activation of Piezo1, a mechanosensitive ion channel, which subsequently induces the nuclear translocation of YAP [38].

### 4.4. Pathological Stretch and Tissue Tension

In vivo and in vitro, adherent cells experience mechanical forces, with stretch being a primary form. Pathological stretch, cyclic or static, promotes myofibroblast transdifferentiation and immune activation via mechano-sensitive ion channels [39]. During tissue growth or disease, ECM stiffening generates static tension that stretches cells, regulating their behavior through YAP/TAZ activity [40]. External stretch forces also modulate cell functions through the Hippo pathway. A recent study has linked YAP dysregulation to cardiovascular dysfunction from PO, highlighting the key role of the Hippo-YAP/TAZ pathway in cellular stretch responses [41].

Mechanistically, this tension stress enhances F-actin polymerization and stretching, increasing nuclear envelope tension, enlarging nuclear pores, and facilitating YAP nuclear entry. This upregulates proliferation-related genes. Concurrently, mitochondria are transported to the perinuclear region along newly formed microtubules, where they produce ATP to support YAP nuclear translocation and partially fuel F-actin polymerization [42]. Notably, cyclic stretching activates YAP across the cytoskeletal-to-nuclear space, leading to epigenetic alterations like H3K9 methylation downregulation [43]. Interestingly, mechanical stretching has an anisotropic effect, with stretch parallel to the cell’s long axis activating YAP nuclear translocation, but perpendicular stretch does not [44]. In summary, mechanical forces, particularly stretch, play a pivotal role in regulating cellular behavior through the Hippo-YAP/TAZ pathway, influencing proliferation, differentiation, and epigenetic modifications.

## 5. YAP/TAZ: Bridging the Interrelationship Between Diabetic Cardiomyopathy and Mechanotransduction

Cells and tissues sense physical forces originating from both internal and external microenvironments. Internally, these forces are primarily generated by cytoskeletal dynamics, termed endogenous forces, whereas external forces stem from the surrounding environment and neighboring cells [40]. In DCM, a significant portion of these external forces arises from the pathologically remodeled ECM, which is in direct contact with cardiomyocytes. The subcellular localization and activity of YAP are modulated by substrate rigidity, and its activation within cardiac fibroblasts (CFs) stimulates ECM stiffening via enhanced collagen deposition [45]. Consequently, the stiffened ECM induces tension in these cells, triggering the high expression and nuclear translocation of YAP [46]. This, in turn, may establish a vicious cycle of “YAP activation-ECM stiffening-mechanical transduction” in DCM.

### 5.1. Cytoskeleton Protein

The cytoskeleton, composed of microtubules, filamentous actin (F-actin), and intermediate filaments (IFs), plays a pivotal role in ensuring the cell’s ability to respond to and propagate signals emanating from both the environment and intracellular forces [47]. This three-dimensional network of filamentous proteins interconnects with cytoplasmic proteins, membranes, organelles, and, notably, the nucleus, serving as a mediator of force transmission [48]. Through the cooperative interactions of microfilaments and microtubules, mechanical signals are efficiently relayed from the cytoskeleton to the nucleus [49]. Furthermore, the dynamics of the cytoskeleton are capable of transmitting mechanical stress to the nuclear envelope, thereby facilitating YAP nuclear translocation. These findings provide further insight into the signaling pathway by which mechanical cues are transmitted from the plasma membrane, through the cytoplasm, and ultimately to the nucleus, ultimately shaping cell fate decisions.

Cell cytoskeleton damage leads to an increase in cardiomyocyte stiffness, and such changes are manifested in diabetic cardiomyocytes [50]. Specifically, diabetic cardiomyocytes show a notable increase in elastic modulus and altered actin organization, with diffuse, irregular deposition and disordered, fragmented filaments [51]. At the onset of DCM, immunocytochemistry showed elevated f-actin levels due to glucose overload [52]. However, excessive F-actin accumulation disrupts sarcomeric structure in cardiomyocytes, impairing cardiac contractility, while enhanced actin polymerization leads to uncoordinated contraction due to heterogeneous Ca^2+^ release [53]. YAP activation is contingent upon F-actin stress fiber-mediated nuclear pore opening. Lee, Joanna Y et al. [54] used co-IP/MS to identify β1 and β4 integrins as key proteins for YAP translocation, bridging the cytoskeleton and ECM. YAP preferentially binds active β1 and inactive β4 integrins, influencing cell area and mediating nuclear translocation. Studies have shown that F-actin regulation plays a crucial role in the mechanical activation of YAP/TAZ. Mechanical stretch or ECM rigidity promotes F-actin polymerization, facilitating nuclear pore expansion and YAP nuclear entry [55]. Inhibition of F-actin polymerization disrupts this process, impairing YAP activation [56].

In DCM, AGEs induce a more stable state of microtubules, manifested by an increase in microtubule protein density and elevated levels of microtubule acetylation [57]. This stable state enhances the stiffness and viscoelasticity of the heart during diastole in HF [58]. Microtubules regulate pathways, sensing and transmitting mechanical stimuli [59]. Acetylated microtubules promote YAP nuclear translocation even under low stiffness. In fibroblasts with low acetylation, YAP localizes to the cytoplasm [60]. Furthermore, stretch reduces microtubule fragmentation, enhances stability, and facilitates YAP nuclear translocation [61]. IFs, while less crucial than actin or microtubules for cell/tissue mechanics, shape cells, direct organelle placement, and regulate interactions [47]. Their unique properties endow cells with resistance against mechanical stresses, maintain structural integrity under extreme strain, stiffen upon compression, and function as springs/shock absorbers in soft tissues [62]. In DCM, nitrosative stress disrupts the structure, promoting myocardial collagen deposition, but metformin can alleviate this dysregulation [63]. High glucose induces the rearrangement and aggregation of IFs, leading to endothelial inflammation and dysfunction [64]. Notably, IFs interact with microfilaments and microtubules to stabilize the cytoskeleton, enhancing cellular mechanosensitivity, and are associated with increased YAP/TAZ nuclear localization post-mechanical stimulation, thereby facilitating mechano-regulation of cell fate [65].

### 5.2. Extracellular Matrix Remodeling

ECM remodeling, a pivotal pathological hallmark of DCM, is instigated by oxidative stress, inflammation, and diverse metabolic disturbances within the heart [66]. This remodeling process directly impacts the mechanical microenvironment of cardiomyocytes, exacerbating myocardial damage and dysfunction in DCM. One of the key consequences of ECM remodeling in DCM is increased ECM stiffness, which stems from multiple factors. Central to this process are CFs, the primary cells responsible for producing ECM proteins and affecting ECM remodeling. In response to profibrotic signals, CFs are activated and differentiate into myofibroblasts [67]. Under high-glucose conditions, this phenotypic transition is accelerated, leading to excessive production of ECM proteins, particularly collagen [68]. The chronic overexpression of collagen-induced by hyperglycemia results in increased ECM stiffness, which constantly stimulates cardiomyocytes and profoundly affects cardiac function and structure [69]. Importantly, myocardial stiffness is determined not only by the total collagen content but also by the degree of collagen cross-linking [70]. Hyperglycemia fosters AGE accumulation, enhancing collagen cross-linking and stimulating fibroblasts [71]. This ECM stiffening, driven by AGE-accelerated cross-linking, leads to myocardial stiffness and diastolic dysfunction in diabetes mellitus.

ECM stiffness is now recognized as a causal factor in aberrant cell behaviors in DCM, not just a pathological consequence. The mechanical properties of the ECM, especially stiffness, profoundly affect cell activity. Traditionally, the focus has been on how a stiff ECM enhances cardiac fibroblast proliferation via YAP activation [72], with matrix stiffness-induced nuclear deformation further regulating gene expression [73]. Beyond stiffness differences, dynamic changes in ECM stiffness also regulate cell behavior. Static stiff matrices promote MSC paracrine function through YAP activation induced by F-actin polymerization [74], but dynamic matrices with time-varying stiffness exhibit stronger YAP activation, leading to increased ECM secretion and gene expression [75]. The duration of preconditioning on soft matrices crucially regulates YAP activity, with prolonged pretreatment sustaining YAP deactivation even on subsequent stiff matrices [76]. This suggests cells retain a mechanical memory, influencing their fate upon transitioning to stiff environments.

Physiologically, the ECM exhibits appropriate viscoelasticity to compensate for mechanical stimuli, with significant energy dissipation. Contrary to current models, substrate viscous energy dissipation diminishes mechanosensing, reducing YAP nuclear translocation and cell spreading [77], while the abnormal matrix viscoelasticity induces YAP nuclear translocation and epithelial-to-mesenchymal transition [78]. In type 2 diabetes mellitus, the ECM accumulates AGEs, altering collagen architecture and enhancing viscoelasticity without affecting stiffness. Enhanced viscoelasticity promotes YAP activation via the integrin-YAP signaling pathway in animal and 3D cell culture studies [28].

## 6. YAP/TAZ in the Pathological Mechanisms of Diabetic Cardiomyopathy

YAP and TAZ, the effectors of the Hippo pathway, serve as mechanotransducers that relay cytoskeletal tension to the nuclei and regulate cellular functions. Furthermore, they also act as mechanical sensors that respond to external stresses [40]. More importantly, they are functionally indispensable for the biological outputs triggered by mechanical cues. In the context of diabetic cardiomyocytes, the response to mechanical stress in the surrounding microenvironment further stimulates the expression and nuclear translocation of YAP and TAZ, which in turn exacerbates cardiac pathological changes and accelerates the progression of diabetic cardiomyopathy (Figure 3).

### 6.1. Cardiac Fibrosis

Hyperglycemia-induced cardiac fibrosis is a prominent feature of DCM, characterized by the deposition of ECM proteins in the cardiac interstitium. Activated fibroblasts and myofibroblasts serve as the central cellular effectors in this process of cardiac fibrosis [79]. In parallel, YAP is also regulated by blood glucose levels, promoting the progression of fibrosis. Compared with the control group, the expression levels of YAP protein in rat myocardial tissues were significantly elevated at both 2 and 4 months post-streptozotocin (STZ) injection. This upregulation further exacerbated fibrosis in both myocardial tissues and CFs under high-glucose conditions [80,81]. Moreover, in diabetic mice, elevated YAP expression disrupted autophagy, subsequently triggering the activation of myocardial fibroblasts. This activation led to enhanced cell proliferation, increased collagen production, and upregulated α-smooth muscle actin (α-SMA) expression [82]. As previously mentioned, these changes contribute to increased ECM stiffness, inducing mechanical transduction, which further exacerbates the pathological process [69,70]. Through the employment of mechanically regulated in vitro models, evidence was provided that a stiff ECM promotes elevated YAP levels and nuclear accumulation within CFs, concomitant with enhanced cellular activation [46].

The mechanical hallmarks of fibrotic microenvironments are both consequences and drivers of fibrosis progression. These hallmarks involve the evolution of intracellular mechanotransduction pathways in response to ECM remodeling and prevalent aberrant hemodynamic conditions in hearts, including low and disturbed shear stress, pathological stretch, and elevated pressure [83]. Pharmacological inhibition of YAP expression and nuclear translocation effectively delay the progression of fibrosis induced by pathological matrix stiffness [84]. This suggests that by inhibiting the expression and nuclear translocation of YAP, cardiac fibrosis in DCM can be ameliorated to some extent, indicating that YAP may represent a novel therapeutic target for this condition.

### 6.2. Cardiac Hypertrophy

Cardiac hypertrophy, a hallmark of various heart diseases, including DCM, is induced by high glucose-mediated myocardial energetic imbalance [3]. In DCM, the myocardium often exhibits a hypertrophic phenotype, characterized by an increase in cardiomyocyte cross-sectional area and elevated expression of hypertrophy-associated markers [85]. Under high glucose stress, cells enlarge (without proliferation) as an adaptive response, resulting in increased myocyte length (eccentric hypertrophy) or width (concentric hypertrophy) and, subsequently, ventricular wall thickening [86].

Pathological cardiac hypertrophy is accompanied by the activation of YAP, which serves as a link between mechanical stimuli and hypertrophy. Specifically, cardiomyocyte-specific deletion of Yap mitigates hypertrophy induced by chronic mechanical stress overload, whereas overexpression of YAP is sufficient to recapitulate the phenotype of such stress-induced hypertrophy [87]. Additionally, PO promotes the accumulation of glycolytic metabolites, including l-serine, l-aspartate, and malate, through the interaction of YAP with TEAD1, ultimately leading to cardiac hypertrophy [88]. Furthermore, transient receptor potential vanilloid 4 (TRPV4), a mechanosensitive channel, has been implicated as a key mediator of matrix stiffness and other mechanical stimuli. Deletion of endothelial TRPV4 protects the heart from PO-induced hypertrophy [89]. Importantly, TRPV4 is crucial for the nuclear translocation of YAP/TAZ in response to matrix stiffness, and its deletion inhibits the matrix stiffness-induced expression of YAP/TAZ proteins [90]. Diabetes mellitus is rarely the sole cause of cardiac dysfunction and often coexists with hypertension; thus, emphasizing blood pressure control in DCM is crucial to alleviate, to some extent, the myocardial mechanical stress induced by PO.

### 6.3. Myocardial Inflammation

Accumulating evidence suggests that chronic myocardial inflammation is not only associated with DCM but also contributes to its development [91]. Clinically, diabetes mellitus and metabolic syndrome patients show elevated inflammatory markers [92,93]. In vitro, high-glucose treatment increases cardiac inflammation in fibroblasts, with higher IL-1β and NF-κB activity [94]. Myocardial biopsy specimens from HF patients with diabetes mellitus reveal increased YAP expression and nuclear localization in cardiomyocytes. The combination of an HFD and PO augmented myocardial leukocyte, macrophage, and neutrophil infiltration associated with YAP activation, which is attenuated by the YAP inhibitor verteporfin [21].

Macrophages, key cells in the innate immune system, are essential for pathogen defense, development, homeostasis, and tissue repair. They clear pathogens, apoptotic cells, and debris, regulate inflammation, and contribute to wound healing [95]. In high glucose conditions, macrophages adopt an M1 pro-inflammatory phenotype, with increased TNF-α, CCL2, and IL-6 [68]. During DCM development, M1 polarization of macrophages is crucial for oxidative stress and ECM formation, leading to myocardial fibrosis via IL-1β secretion [96].

YAP/TAZ is demonstrated as a key regulator of macrophage polarization and function, acting as an activator in pro-inflammatory macrophages and a repressor in reparative ones. Specifically, YAP/TAZ promotes a pro-inflammatory phenotype by regulating the IL6 promoter or the p38-MAPK pathway [97]. Active YAP enhances inflammatory activation, and the macrophage inflammatory response is modulated by the biophysical microenvironment via YAP. In vivo, soft materials decrease the expression of inflammatory markers and YAP in adjacent macrophages compared to stiff materials. Furthermore, nuclear YAP amplifies macrophage responsiveness to the inflammatory agonist lipopolysaccharide [98]. Influenced by substrate stiffness, YAP promotes an M1 pro-inflammatory phenotype on stiff matrices through mechanotransduction signaling, whereas it favors an M2 anti-inflammatory phenotype on soft and medium stiffness substrates [99].

### 6.4. Programmed Cell Death

Programmed cell death (PCD), a genetically regulated process encompassing apoptosis, autophagy, necroptosis, ferroptosis, and pyroptosis, is characterized by aberrant pathway activation that is widespread in nature [100]. This dysregulation contributes to excessive cardiac remodeling and plays a pivotal role in the pathogenesis of numerous cardiovascular diseases, ultimately culminating in HF [101]. Diabetes mellitus induces PCD in the myocardium, consequently diminishing cardiac contractility and predisposing individuals to HF and other related complications [102].

High glucose stimulation exacerbates DCM progression by aggravating mitochondrial damage, enhancing ROS accumulation, promoting necroptosis, and activating the NLRP3 inflammasome [103]. In vitro, high glucose and palmitate acid induce oxidative stress and apoptosis in cardiomyocytes, consistent with observations in DCM model mice [104]. Previously, the Hippo pathway’s activation led to increased YAP phosphorylation, causing its degradation in the cytoplasm and exacerbating cardiomyocyte necroptosis [105]. YAP phosphorylation also induces apoptosis in mouse liver tissues [106]. However, YAP translocates to the nucleus, interacting with TEAD transcription factors to stimulate anti-apoptosis and proliferation genes [107]. Nevertheless, during apoptosis, tissue relaxation and mechanical stretching of neighboring cells occur, with greater stretch in nearest neighbors. This stretching, due to tissue relaxation and actomyosin contraction, links to spatial heterogeneity in cell division and YAP nuclear translocation [108]. Consequently, cells near apoptotic cells are more likely to undergo YAP nuclear translocation, potentially exacerbating myocardial fibrosis and inflammation in DCM.

Unlike necrosis and apoptosis, YAP upregulation and nuclear translocation induce ferroptosis. Research indicates that ferroptosis is linked to cell death induced by stiff substrates via mechanically mediated YAP nuclear translocation [109]. Additionally, ferroptosis occurs in cardiac microvascular ECs post-diabetes, with suppressed Pink1/Parkin-dependent mitophagy [110]. Furthermore, AGEs are capable of inducing ferroptosis in engineered cardiac tissue (ECT) as well as in mice with DCM [111]. Wang, Wang et al. [112] revealed that high glucose and high-fat conditions upregulate YAP expression and activate ACSL4, a key regulator of ferroptosis, thereby accelerating ferroptosis.

### 6.5. Mitochondrial Dysfunction

Mitochondrial dysfunction plays a crucial role in either inducing or exacerbating oxidative stress in the diabetic heart. In these hearts, mitochondrial fission is overactivated, resulting in an increased number of mitochondria within cardiomyocytes, whereas physiological fusion is markedly inhibited [113,114]. Under normal physiological conditions, damaged mitochondria undergo mitophagy; however, this process becomes impaired under hyperglycemic conditions, leading to the accumulation of non-functional mitochondria, which may ultimately induce cardiomyocyte death [115,116].

Interestingly, in cardiomyocytes overexpressing YAP, a reduction in mitochondrial cross-sectional area has been observed, suggesting the occurrence of mitochondrial hyper-fragmentation. This phenomenon is particularly evident in the context of cardiac hypertrophy induced by chronic mechanical stress overload [87]. Mechanical cues, such as ECM stiffness, spatial confinements, and applied forces like stretching, play a pivotal role in regulating mitochondrial dynamics. These mechanical transductions stimulate the expression and nuclear translocation of YAP, which in turn are associated with the recruitment of dynamin-related protein 1 (DRP1) to the mitochondria. This recruitment influences peri-mitochondrial F-actin formation and subsequently promotes mitochondrial fission [117].

Conversely, the suppression of YAP’s nuclear translocation leads to a decrease in DRP1 synthesis, thereby inhibiting mitochondrial fission and also reducing the accumulation of lipid droplets and cholesterol [118]. Furthermore, the deletion of Yap results in the collapse of F-actin, manifesting as an unclear, disorganized, and fragmented state. This disruption subsequently triggers the activation of mitophagy [119]. Nevertheless, further studies are warranted to elucidate the direct role of YAP/TAZ in mitochondrial homeostasis in DCM.

## 7. YAP/TAZ: A Promising Therapeutic Target for Diabetic Cardiomyopathy

Several inhibitors targeting YAP/TAZ have been explored for their potential therapeutic applications in DCM. Verteporfin, a drug that modulates the activity of the YAP/TAZ complex by disrupting its interaction with transcriptional co-activators, has shown potential in regulating relevant cellular processes. This disruption subsequently inhibits downstream gene expression regulated by YAP/TAZ [120]. It has shown promise in reducing cardiomyocyte oxidative stress and fibrosis in DCM models [80]. Another inhibitor, JQ1, a bromodomain and extraterminal protein inhibitor, was found to downregulate YAP transcription and directly repress YAP, TAZ and TEAD expressions [121]. Mu J et al. [122] discovered that JQ1 treatment in DCM mice enhanced mitochondrial function and restored the structural and functional integrity of the diabetic heart. Additionally, the administration of CA3, a recently developed YAP chemical inhibitor, remarkably attenuated TGF-β-mediated myofibroblast differentiation and collagen production [123]. Nevertheless, the therapeutic efficacy of CA3 in DCM remains to be further explored. These inhibitors targeting YAP/TAZ offer different mechanisms and show various degrees of potential in treating DCM, but more research is needed to fully understand their clinical applications.

In addition, certain medications used for treating diabetes mellitus and cardiovascular diseases may also exert their therapeutic effects by regulating YAP and TAZ. SGLT2 inhibitors, a class of antidiabetic drugs, have been proven to have cardioprotective effects in addition to their role in glycemic control [124]. A recent study has suggested that the SGLT2 inhibitor dapagliflozin may modulate the YAP/TAZ pathway [125]. Both in vitro and in vivo experiments demonstrated that dapagliflozin, similar to verteporfin, reduces YAP/TAZ activation and downregulates the expression of target genes, such as connective tissue growth factor. However, the precise molecular mechanisms underlying the interaction between SGLT2 inhibitors and YAP/TAZ require further investigation. Moreover, metformin, a first-line drug for type 2 diabetes mellitus, has been associated with potential beneficial effects on the heart. Emerging evidence indicates that metformin can inhibit endothelial YAP, which effectively restores endothelial function in obese and diabetic conditions [126]. Statins, commonly employed as lipid-lowering agents for cardiovascular disease prevention, have also demonstrated potential regulatory effects on YAP/TAZ. Through a high-throughput small-molecule screen in primary human lung fibroblasts, multiple statins were discovered to inhibit YAP nuclear localization by inducing YAP phosphorylation, leading to cytoplasmic retention and subsequent degradation [127].

In conclusion, the YAP/TAZ signaling pathway represents a highly promising therapeutic target for DCM. Multiple medications with different primary indications have shown the potential to interact with this pathway, suggesting that modulating YAP/TAZ could be a novel and effective strategy for treating DCM. Nevertheless, comprehensive investigations are essential to elucidate the precise molecular mechanisms, optimize treatment regimens, and translate these findings into clinical practice for better patient outcomes.

## 8. Conclusions and Prospect

DCM is characterized by a constellation of symptoms, including myocardial fibrosis, hypertrophy, cardiac stiffness, and diastolic dysfunction, which may ultimately progress to systolic dysfunction and clinical HF [128]. Although the condition features an extensive preclinical course, a precise definition of DCM as a distinct clinical entity remains challenging. This ambiguity stems from the lack of universally accepted diagnostic criteria and the scarcity of information on subclinical cardiovascular disease in the early stages of diabetes mellitus. Moreover, there is no consensus on the optimal management strategy for its prevention or treatment, and no standardized pharmacological therapy for DCM is currently available. Despite these challenges, the volume of related research is, encouragingly, on the rise each year [129].

The manifestation of cardiac stiffness and impaired cardiac function progressively sculpts a detrimental mechanical environment for the heart, exacerbating pathological cardiac remodeling. From a mechanistic perspective, aberrant mechanical cues, including ECM stiffness, shear stress, disturbed flow, pathological stretch, and pressure, are implicated in the pathophysiological processes underlying cardiac inflammation, mitochondrial dysfunction, oxidative stress, and cell death. Underlying these pathophysiological events, YAP and TAZ, the effectors of the Hippo pathway, serve as mechanotransducers that transmit these mechanical forces through the cytoskeleton to the nucleus, thereby regulating cellular functions. To date, attention to mechanotransduction in DCM has been insufficient, and further exploration is warranted to disrupt the vicious cycle of cardiac mechanical remodeling in DCM.

## Figures and Tables

**Figure 1 cimb-47-00297-f001:**
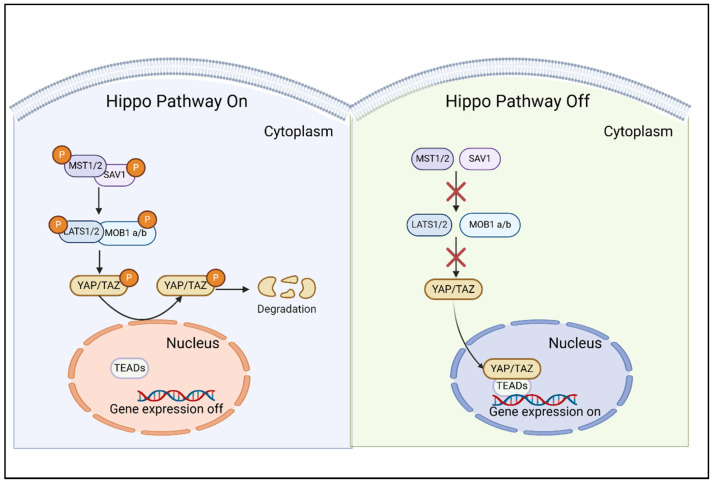
**Overview of the Hippo-YAP pathway.** Upon activation of the Hippo signaling pathway, the downstream Hippo effectors YAP/TAZ are phosphorylated by the Hippo core kinases Lats1/2, leading to their retention in the cytoplasm and subsequent degradation. Conversely, when the Hippo signaling is inactivated, YAP/TAZ functions as transcriptional cofactors that translocate into the nucleus and interact with other transcription factors, such as members of the TEA domain transcription factor family (TEADs), to regulate gene expression. Created in BioRender. SHEN, J. (2025) https://BioRender.com/b50g901 (accessed on 18 March 2025).

**Figure 2 cimb-47-00297-f002:**
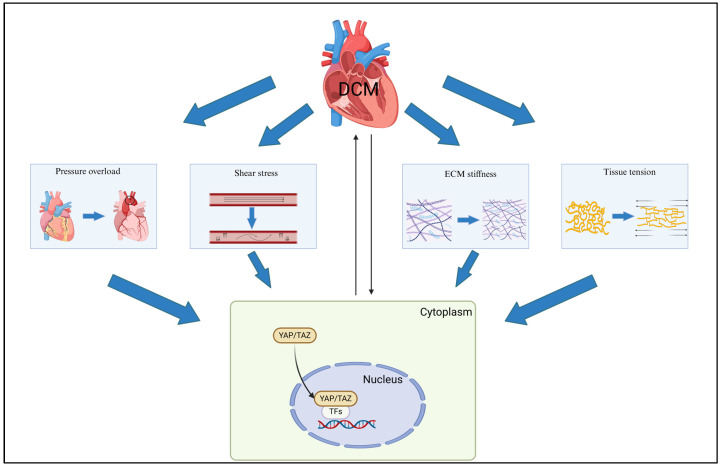
**YAP/TAZ serves as a link between DCM and mechanical stimulus signaling.** In diabetic hearts, mechanical stimuli such as pressure overload, shear stress, extracellular matrix (ECM) stiffness, and tissue tension elicit responses, with these signals being transmitted through complex pathways to YAP/TAZ in the cytoplasm. This promotes YAP/TAZ expression and nuclear translocation, subsequently regulating transcription factors (TFs) in the nucleus and ultimately leading to alterations in cardiac structure and function. Created in BioRender. SHEN, J. (2025) https://BioRender.com/n19k971 (accessed on 18 March 2025).

**Figure 3 cimb-47-00297-f003:**
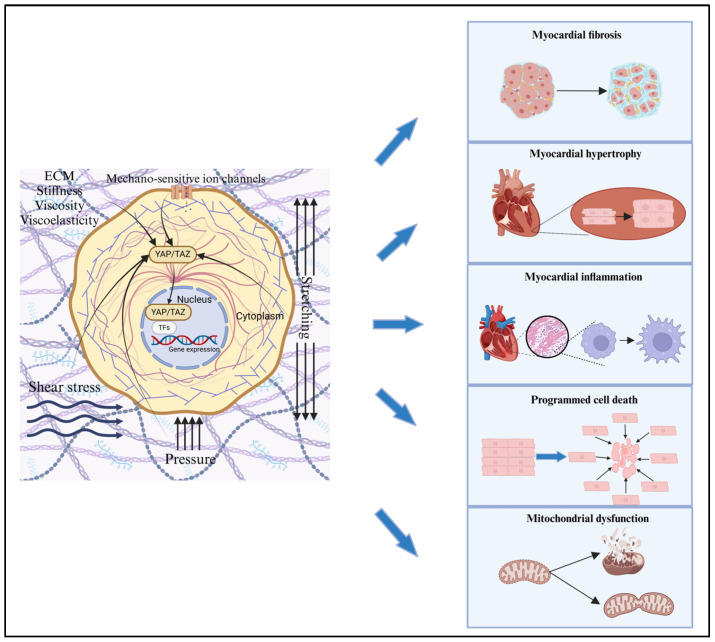
**The schematic diagram of YAP/TAZ mediating cardiac pathological changes.** In diabetic cardiomyopathy, YAP/TAZ plays a crucial role in mediating mechanical signals such as ECM stiffness, shear stress, pressure overload, and stretching. These mechanical cues are transmitted to the nucleus via the cytoskeleton, resulting in increased YAP/TAZ expression and nuclear translocation. This, in turn, triggers downstream gene expression, which contributes to the progression of myocardial fibrosis, hypertrophy, inflammation, programmed cell death, and mitochondrial dysfunction. Created in BioRender. SHEN, J. (2025) https://BioRender.com/e48g548 (accessed on 18 March 2025).

## Data Availability

No datasets were generated or analyzed during the current study.

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
