# Peer review of "The Mechanical Role of YAP/TAZ in the Development of Diabetic Cardiomyopathy"

_cimb, 2025, doi:10.3390/cimb47050297_

Round 1

Reviewer 1 Report

Comments and Suggestions for Authors

This a very comprehensive review of YAP/TAZ signaling in Diabetic Cardiomyopathy and Mechanotransduction. The pathological role of YAP/TAZ signaling in pathological mechanism of cardiomyopathy has been reviewed comprehensively in addition to the role of YAP/TAZ in. normal cell mechano transduction. YAP/TAZ signaling has been studied extensively in fibrotic diseases but its role in diabetic cardiomyopathy needed much needed comprehensive literature review. This review does justice to that aspect of the YAP/TAZ signaling pathway.

Author Response

Comments 1: This a very comprehensive review of YAP/TAZ signaling in Diabetic Cardiomyopathy and Mechanotransduction. The pathological role of YAP/TAZ signaling in pathological mechanism of cardiomyopathy has been reviewed comprehensively in addition to the role of YAP/TAZ in. normal cell mechano transduction. YAP/TAZ signaling has been studied extensively in fibrotic diseases but its role in diabetic cardiomyopathy needed much needed comprehensive literature review. This review does justice to that aspect of the YAP/TAZ signaling pathway.
Response 1:  Thank you very much for your meticulous review and high praise of our review paper! Your recognition is extremely encouraging and has instilled greater confidence in our research work. When writing this paper, we were keenly aware that although there have been some studies on the YAP/TAZ signaling pathway in the field of diabetic cardiomyopathy, a systematic summary was lacking, especially regarding its association with mechanotransduction. Therefore, we conducted an extensive literature reviews, striving to comprehensively and thoroughly elucidate the roles of YAP/TAZ in normal cellular mechanotransduction and the pathological mechanisms of diabetic cardiomyopathy. We are delighted that our efforts have received your approval, which indicates that our research direction and content have met the expected goals.

Reviewer 2 Report

Comments and Suggestions for Authors

In the paper entitled, “The Mechanical Role of YAP/TAZ in the Development of Diabetic Cardiomyopathy”, Shen J-X et al reviewed recent progress on the crucial role of the Yes-associated protein and its transcriptional coactivator with PDZ-binding
motif (YAP/TAZ) mechanotransduction in the pathological
changes observed in diabetic cardiomyopathy.
The review is well structured and details the main pathological processes in diabetic cardiomyopathy in the potential relationship with YAP/TAZ.
To add value to this review, I suggest the authors to write a short paragraph about known or under-study inhibitors of YAP/TAZ or about other potential therapeutic targets in these pathological processes. Furthermore, I would suggest a brief description of the influence of antidiabetic medication, especially SGLT2 inhibitors, on the YAP/TAZ mechanism.
The rest are small grammatical errors that should be corrected, such as the penultimate line in the conclusions section which begins with a word that is missing a letter.

Author Response

Comments 1: In the paper entitled, “The Mechanical Role of YAP/TAZ in the Development of Diabetic Cardiomyopathy”, Shen J-X et al reviewed recent progress on the crucial role of the Yes-associated protein and its transcriptional coactivator with PDZ-binding motif (YAP/TAZ) mechanotransduction in the pathological changes observed in diabetic cardiomyopathy. The review is well structured and details the main pathological processes in diabetic cardiomyopathy in the potential relationship with YAP/TAZ. To add value to this review, I suggest the authors to write a short paragraph about known or under-study inhibitors of YAP/TAZ or about other potential therapeutic targets in these pathological processes. Furthermore, I would suggest a brief description of the influence of antidiabetic medication, especially SGLT2 inhibitors, on the YAP/TAZ mechanism.
Response 1: Thank you for your constructive suggestions. We highly appreciate your insights, which have significantly enhanced the quality and comprehensiveness of our review. In response to your recommendations, we have added a dedicated section (Section 7, pages 12-13) that explores both known and ongoing research on inhibitors of the YAP/TAZ pathway, as well as other potential therapeutic targets in the pathological processes of diabetic cardiomyopathy (DCM). We have detailed the mechanisms of action of several YAP/TAZ inhibitors, including verteporfin, JQ1, and CA3, and their effects on DCM in pre-clinical models. Additionally, we have expanded on the influence of antidiabetic medications, particularly SGLT2 inhibitors, on the YAP/TAZ mechanism. We also included discussions on metformin and statins, demonstrating how these widely-used drugs may regulate YAP/TAZ signaling and contribute to cardioprotection in DCM.

Comments 2: The rest are small grammatical errors that should be corrected, such as the penultimate line in the conclusions section which begins with a word that is missing a letter.

Response 2: Thank you for meticulously identifying the small grammatical errors in our manuscript. We sincerely appreciate your careful review and valuable feedback. The error you mentioned, where a letter was missing from the initial word of the penultimate line, has been promptly corrected (Page13).